# Genomic Differences and Mutations in Epidemic Orf Virus and Vaccine Strains: Implications for Improving Orf Virus Vaccines

**DOI:** 10.3390/vetsci11120617

**Published:** 2024-12-02

**Authors:** Dengshuai Zhao, Yaoxu Shi, Miaomiao Zhang, Ping Li, Yuanhang Zhang, Tianyu Wang, Dixi Yu, Keshan Zhang

**Affiliations:** 1Guangdong Provincial Key Laboratory of Animal Molecular Design and Precise Breeding, College of Animal Science and Technology, Foshan University, Foshan 528225, China; zds523072053@163.com (D.Z.); 13526707484@163.com (M.Z.); liping00716@163.com (P.L.); zyhh105@163.com (Y.Z.); tianyuwang0324@163.com (T.W.); 15940297229@163.com (D.Y.); 2College of Veterinary Medicine, Henan University of Animal Husbandry and Economy, Zhengzhou 450046, China; shiyaoxu@163.com

**Keywords:** Orf virus, Orf attenuated vaccine strain, whole genome sequencing, phylogenetic analysis, sheep

## Abstract

The effectiveness of the live attenuated Orf virus vaccine is influenced by several factors, including the genomic match between the vaccine strain and circulating epidemic strains. Genome-wide analysis showed a high similarity between the vaccine and epidemic strains, though single nucleotide mutations were found in *ORF067*, *ORF072*, *ORF102*, and the terminal non-coding region. These mutations likely result from heterologous transmission of the vaccine strain. Although the vaccine provides effective immunity, these variations highlight the need for ongoing monitoring of viral evolution. Identifying these mutations may help improve vaccine design and reveal potential virulence genes.

## 1. Introduction

Orf (ORF) is a zoonotic disease, primarily affecting goats and sheep, though rarely seen in humans [1]. The key clinical symptoms in sheep infected with Orf virus (ORFV) include erythema on the lips, nose, oral mucosa, tongue, and udder, which progress to blister formation, pustules, and eventually scabs [2,3]. Secondary infections are a major cause of high mortality in lambs with ORFV, occurring in up to 93% of cases [4]. ORFV is globally distributed and causes significant economic losses to the sheep industry [5]. It is a large, double-stranded DNA virus belonging to the genus *Parapoxvirus* within the family *Poxviridae* [6]. The ORFV genome is approximately 140 kb in length and encodes 132 genes [7]. The genome consists of a relatively conserved central region and highly variable terminal regions. The central region, rich in G+C content, is mainly involved in viral replication, while the genes in the terminal regions are primarily responsible for pathogenicity [8,9].

Vaccination is an effective and cost-efficient method for preventing ORFV infection [10]. However, in vivo studies have shown that immunization with inactivated ORFV does not prevent reinfection in hosts [11]. Attenuated vaccines provide sufficient protection against ORFV, and although the protection is not lifelong, annual vaccination is considered the most effective strategy for disease control in flocks and herds [12]. A significant concern with current attenuated vaccines is the risk of virulence reversion. Genes and proteins involved in viral virulence and pathogenic mechanisms may affect the safety and efficacy of existing vaccines against ORFV infection [10].

Therefore, in this study, we identified differences between the whole genomes of an epidemic ORFV strain (ORFV-2W) and a vaccine strain (ORFV-1V) through multiple sequence alignments, phylogenetic analyses, single nucleotide polymorphism (SNP) identification, and protein tertiary structure predictions. This comprehensive genomic analysis may help identify novel virulence gene mutations and guide the future development and optimization of ORFV vaccines.

## 2. Materials and Methods

### 2.1. Samples

The epidemic Orf virus strain (ORFV-2W) was isolated and preserved by the Guangdong Provincial Key Laboratory of Animal Molecular Design and Precise Breeding at Foshan University, and propagated in goat skin fibroblasts cells (GSF). The attenuated Orf vaccine strain GO-BT (ORFV-1V) was obtained from Shandong Tai Feng Biological Products Co., Ltd., (Jinan, China), and propagated in bovine testicular cells.

ORFV-1V was diluted to 1 mL with phosphate-buffered saline (PBS), filtered through a 0.45 μm membrane, and centrifuged at 1000× *g* for 1 min. The supernatant was inoculated into bovine testicular cells at 37 °C in 5% CO_2_ and were blind-passaged three times. Cell cultures were then collected, and the DNA of ORFV-1V and ORFV-2W was extracted using the DNA extraction kit (QIAGEN, Hilden, Germany, 69504). The extracted DNA was confirmed via agarose gel electrophoresis using the B2L gene as a reference fragment. Based on the whole genome sequence of the ORFV strain SJ1 (KP010356.1) published in GenBank, primers targeting the B2L gene (1137 bp) were designed and synthesized using Primer Premier 5 software (Forward: 5′-GATACCACAAGATCAGCCGT-3′; Reverse: 5′-CTGCTCATGGTATCAATCTTATCGA-3′).

### 2.2. Samples DNA Sequencing, Assembly, and Annotation

The genomic DNA of ORFV samples was sequenced using the Illumina Xten platform at Shanghai Zhongsen Biotechnology Co., Ltd., (Shanghai, China). Quality assessment of the sequencing data, both pre- and post-quality control, was conducted using FastQC version 0.11.8. High-quality paired-end reads (~100,150 bp) were assembled into longer contigs through a sequence overlap strategy to generate a draft genome. De novo assembly of the high-quality reads was performed using Shovill version 1.0.9, and contig ends were extended with SSPACE version 3.0. The complete viral genome was obtained by further assembling contigs using CAP3 online tool (https://doua.prabi.fr/software/cap3 accessed on 4 August 2024). Functional annotation of the genome was performed by comparing the sequences of the newly sequenced isolates with the reference genome KP010356.1. The sequence information for this study has been uploaded to GenBank with the accession number PQ374835 and PQ374836.

### 2.3. Sequence Alignment and Phylogenetic Analysis

The nucleotide and amino acid sequences of *ORF020* (VIR) and *ORF127* (vIL-10) from the ORFV-1V and ORFV-2W genomes were compared with those of other ORFV strains available in GenBank. All alignment comparisons were performed using ClustaIW within the Alignment program implemented via MEGA version 11. Additionally, the complete genome nucleotide and amino acid sequences of ORFV-1V and ORFV-2W were compared with the reference ORFV strain SJ1 (KP010356.1). Whole genome sequence alignments of ORFV-1V and ORFV-2W were conducted using EMBOSS online tool (https://www.ebi.ac.uk/jdispatcher/psa/emboss_needle accessed on 4 August 2024). The results of these alignments were visually analyzed with GENEDOC version 2.7. Single nucleotide polymorphisms (SNPs) identified in the sequenced strains were detected using Snippy version 4.4.3, with KP010356.1 as the reference genome.

The complete genome sequences of ORFV-2W, ORFV-1V, and other ORFV strains published in GenBank (Table 1) were analyzed using MEGA version 11 and the neighbor-joining method. A phylogenetic tree was constructed based on two highly conserved genes, *ORF011* (B2L) and *ORF059* (F1L), to explore the genetic relationships between ORFV-1V, ORFV-2W, and reference ORFV strains. Simultaneously, the whole genome sequences of ORFV-1V, ORFV-2W, and other published strains from GenBank were analyzed using a phylogenetic tree. Sequences from 1000 replicates were used to construct the neighbor-joining method, which were generated using MEGA version 11.

### 2.4. Structure Prediction of ORFV Proteins

ORFV proteins were analyzed using the SWISS-MODEL online tool (https://swissmodel.expasy.org/interactive accessed on 4 August 2024) for tertiary structure prediction. Template proteins were selected based on sequencing coverage and the Global Model Quality Estimate (GMQE) of the target-template alignment. Template coverage refers to the extent to which the amino acid sequences of target protein match those of the template protein. GMQE combines factors from both the target-template alignment and structural analysis of the template. It is used prior to constructing the model, aiding in the selection of optimal templates that enhance the reliability of the predicted model. The resulting GMQE score is represented as a value between 0 and 1, with higher values indicating greater reliability and reflecting the expected accuracy of the model based on the selected template.

## 3. Results

### 3.1. Features of the Complete Genomes of the ORFV Epidemic and Attenuated Vaccine Strain

The whole genome length of the vaccine strain (ORFV-1V) was determined to be 130,535 bp with a GC content of 63.60%. The base composition of the ORFV-1V genome is as follows: A (23,894 bp), T (23,619 bp), C (41,358 bp), and G (41,664 bp). The genome length of the epidemic strain (ORFV-2W) was 130,583 bp, with a GC content of 63.56%, and a base composition of A (23,925 bp), T (23,658 bp), C (41,343 bp), and G (41,657 bp). These findings are summarized in Figure 1.

These results indicate that the number of coding sequences (CDSs) in ORFV-1V and ORFV-2W is similar to that in the reference genome KP010356.1. Similar to other ORFV strains, the genomes of ORFV-1V and ORFV-2W feature a large central coding region and non-coding inverted terminal repeats (ITRs).

### 3.2. Genome-Wide Comparison Between the Epidemic ORFV and Attenuated Vaccine Strains

To investigate genome-wide differences between the ORFV-1V and ORFV-2W strains, a multiple sequence alignment analysis was conducted on their complete genomes. The results revealed a sequence homology of 99.8% between ORFV-1V and ORFV-2W. Compared to ORFV-2W, ORFV-1V exhibited a 144-bp deletion in the non-essential terminal region, spanning positions 130,530 bp and 130,674 bp (Figure 2). This deletion in the non-essential terminal regions may suggest reduced virulence, likely due to the in vitro propagation of the ORFV vaccine strain [15], a process the epidemic strain did not undergo.

### 3.3. ITR Analysis of the Epidemic ORFV and Attenuated Vaccine Strains and Other Strains

Similar to other ORFV strains, the terminal regions of the ORFV-1V and ORFV-2W genomes contain inverted terminal repeats (ITRs). The ITRs of ORFV-1V and ORFV-2W were 242 bp and 382 bp in length, respectively. These ITRs were analyzed and compared to those in corresponding regions of other of ORFV strains published in GenBank. We found that the ITRs of ORFV-2W included a conserved telomere resolution sequence at their 5′ end, which was absent in the ITRs of ORFV-1V, similar to the KF234407.1 NA1/11 and KY053526.1 HN3/12 strains. Additionally, the genomic sequence of ORFV-2W was more complete than that of ORFV-1V. Both ORFV-1V and ORFV-2W ITRs contained large deletion regions that were absent in seven other ORFV strains: KP010354.1, KP010356.1, KP010353.1, MN648218.1, MN648219.1, MG674916.2, and MG712417.1 (Figure 3). The ITRs of ORFV-1V had a shorter deletion region compared to ORFV-2W, with the deletion spanning 140 bp, accounting for most of the difference in genome length between the two strains.

### 3.4. Phylogenetic Analysis

B2L and F1L are commonly used in the molecular diagnosis, characterization, and phylogenetic analysis of ORFV. To further elucidate the genetic relationship between ORFV-1V, ORFV-2W, and other ORFV strains, and to validate the results of our multiple sequence alignment analysis, nucleotide sequences corresponding to B2L and F1L from different ORFV strains were obtained from GenBank. These sequences were used to construct a phylogenetic tree in MEGA version 11 using the neighbor-joining method with 1000 replicates. The analysis revealed that ORFV-1V and ORFV-2W clustered in the same branch, indicating a high degree of homology and the closest genetic relationship (Figure 4A,B). Furthermore, a phylogenetic analysis of the complete ORFV genome (Figure 4C) also showed that ORFV-1V and ORFV-2W clustered on the same branch, again demonstrating the highest homology and the closest genetic relationship. These findings are consistent with our single-gene (B2L and F1L) analysis.

### 3.5. SNPs in the ORFV Vaccine Strain Genome

In this study, we analyzed the single nucleotide polymorphisms (SNPs) in the ORFV-1V genome using snippy version 4.4.3. SNPs in the reference genome ORFV strain SJ1 (KP010356.1) were compared with those of the vaccine and epidemic strains. The results showed that the SNPs in ORFV-1V were primarily missense and frameshift mutations, concentrated in *ORF067*, *ORF072*, *ORF102*, as well as in non-essential regions at the genome’s terminal ends. To assess the potential impact of these SNPs, we further examined the nucleotide and amino acid sequences of *ORF067*, *ORF072*, and *ORF102* in ORFV-1V and other strains.

We identified a “G” to “C” point mutation at nucleotide position 586 of *ORF067* in ORFV-1V, which resulted in a substitution mutation at amino acid position 196, changing aspartic (Asp) to histidine (His) (Figure 5A). Additionally, a “T” to “C” SNP at position 337 in *ORF072* led to the mutation of isoleucine (Ile) to valine (Val) at position 113 (Figure 5B). A third SNP, located in the non-coding region at position 111,746 near the end of the genome, involved a “C” to “A” substitution. Finally, we identified an insertion at nucleotide position 442 in *ORF102* of ORFV-1V, where “A” was changed to “AA”, resulting in the mutation of amino acid 148 to asparagine (Asn) (Figure 5C). These findings were consistent with the SNPs predicted by Snippy version 4.4.3 in the reference strain (KP010356.1).

### 3.6. Tertiary Structure Prediction of Proteins Encoded by ORF067, ORF072, and ORF102

Our SNP analysis revealed that *ORF067* and *ORF072* of the ORFV-1V genome contained missense mutations, while ORF102 exhibited a frameshift mutation. These SNPs resulted in corresponding amino acid changes in the proteins encoded by *ORF067*, *ORF072*, and *ORF102*. To assess whether these mutations caused conformational changes in the tertiary structures of the proteins, the deduced amino acid sequences of ORFV-1V proteins were submitted to SWISS-MODEL for tertiary structure prediction. The predicted structures were analyzed using SWISS-PDB-VIEWER version 4.1. The coverage rate of *ORF067* (ORFV-1V) to its template was 93%, with a GMQE of 0.87. Similarly, *ORF072* (ORFV-1V) had 99% coverage and a GMQE of 0.84, whereas *ORF102* (ORFV-1V) showed 51% coverage with a GMQE of 0.2. These results suggest that the conformations of the proteins encoded by *ORF067* and *ORF072* remained largely unchanged after the amino acid mutations. However, the protein encoded by *ORF102* underwent significant changes following the introduction of the frameshift mutation (Figure 6).

## 4. Discussion

Our comparative analysis of genome-wide differences between ORFV-1V and ORFV-2W identified a 144 bp deletion in a non-essential terminal region. Previous studies indicate that such regions are lost during the culture of heterologous cells [21]. Gene recombination in the terminal region can lead to the translocation of genomic fragments, generating new variations in the ORFV genome [22]. We speculate that this deletion may be linked to the attenuated virulence of ORFV-1V, although the underlying mechanism requires further investigation.

The envelope genes *ORF011* and *ORF059* encode the highly immunogenic envelope proteins B2L and F1L, respectively. These genes are often used in the molecular diagnosis, characterization, and phylogenetic analysis of ORFV [23]. In this study, phylogenetic analysis based on the single-gene sequences of B2L or F1L showed that ORFV-1V and ORFV-2W cluster together on the same branch. Furthermore, whole genome phylogenetic analysis also confirmed that ORFV-1V and ORFV-2W also clustered on the same branch. Collectively, these findings strongly suggest that ORFV-1V and ORFV-2W are closely related.

SNP analysis of ORFV-1V revealed missense mutations in *ORF067* and *ORF072*, along with a frameshift mutation in *ORF102*. An insertion at nucleotide position 442 in *ORF102* resulted in a threonine (Thr) substitution at position 148 in ORFV-1V, while a deletion at position 443 in *ORF102* of ORFV-2W led to an asparagine (Asn) substitution. *ORF067* encodes an uracil-DNA glycosidase that functions in DNA repair. In particular, this protein removes uracil by hydrolyzing the N-glycosidic bond between uracil and deoxyribose. Due to its antiviral activity, it plays an important role in viral virulence [24] and is essential for both innate and humoral immunity as well as the pathogenesis of DNA viruses [25]. However, its function in ORFV infections remains unexplored. The protein encoded by *ORF072*, known as transcriptional termination factor NPH-1, is one of the protein factors involved in assembling the multi-subunit viral RNA polymerase (vRNAP), which is responsible for ensuring the transcriptional initiation and expression of viral genes [26]. Studies on the vaccine virus have demonstrated that amino acid mutations in NPH-1 can enhance interferon induction [27]. Further amino acid sequence analysis of *ORF102* revealed that it encodes an A-type inclusion (ATI) protein (Appendix A). ATI proteins typically contain mature virions embedded in a dense matrix, which may aid in the prolonged survival and transmission of infectious viruses in the environment [26]. Additionally, genomic analysis of a highly attenuated virus strain revealed that a highly expressed gene responsible for ATI formation, ORF A26L, had been deleted [28]. Next, we performed structural prediction analysis of the proteins encoded by *ORF067*, *ORF072*, and *ORF102*. Our results indicated that the missense mutations in *ORF067* and *ORF072* did not alter the protein conformation. However, the frameshift mutation in *ORF102* of ORFV-1V may have led to structural changes in its encoded protein. Notably, the target-template protein coverage for *ORF102* was only 51%, which may introduce uncertainty into the prediction results.

In summary, we propose that the frameshift mutation in *ORF102* of ORFV-1V, which altered the amino acid sequence and protein conformation of the ATI, may be associated with the attenuated virulence. Additionally, mutations in *ORF067* and *ORF072* may impact DNA repair and vRNAP formation, contributing to this weakened virulence. However, limited research has been conducted on *ORF067*, *ORF072*, and *ORF102* in ORFV, and further investigations are required to fully understand their roles. Thus, functional analyses provide preliminary insights into their impact on pathogenicity, and additional in vitro and in vivo experiments are needed to validate these findings and clarify their contributions to ORFV pathogenicity. This study supports the hypothesis of genome-level differences between viral strains used in the Chinese commercial live attenuated ORFV vaccine and those that are prevalent, potentially informing future vaccine optimization.

## 5. Conclusions

In conclusion, through whole genome resequencing, we generated genomic datasets and performed phylogenetic analysis, which revealed a close genetic relationship between ORFV-1V and ORFV-2W. Our analysis identified key mutations in the ORFV-1V genome, including missense and frameshift mutations in *ORF067*, *ORF072*, and *ORF102*, as well as mutations in the noncoding terminal regions. Notably, a 140-bp deletion was detected in the ITR region of ORFV-1V. The proteins encoded by these genes are implicated in antiviral immunity and virulence, though their specific roles in ORFV infection remain to be clarified. Therefore, further functional analyses of these proteins and the ITR region are essential for identifying and characterizing novel virulence genes in ORFV. These findings provide valuable insights for the improvement and optimization of commercial ORFV vaccines.

## Figures and Tables

**Figure 1 vetsci-11-00617-f001:**
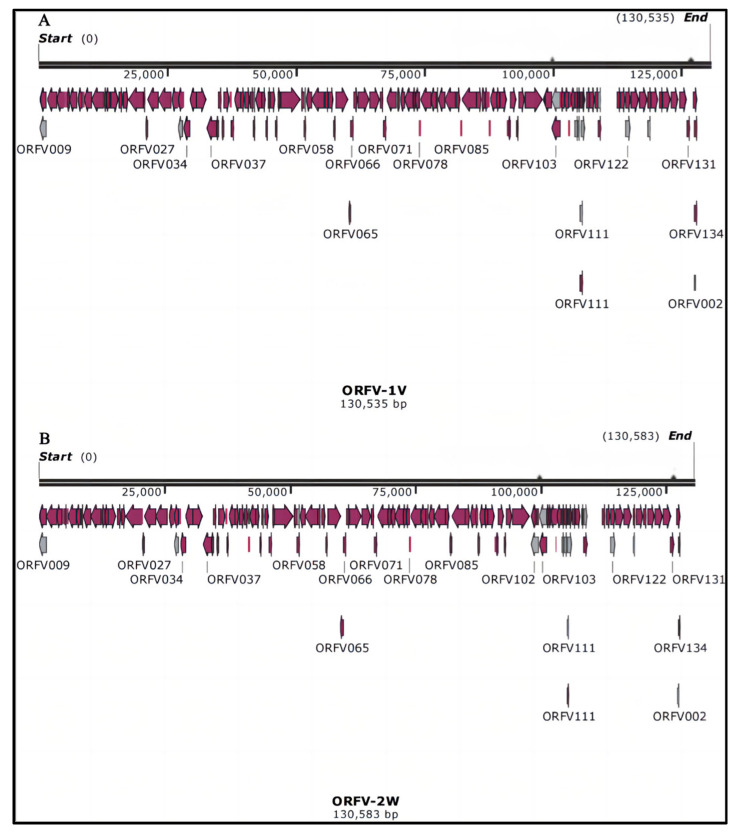
Whole genome maps of ORFV-1V and ORFV-2W. (**A**) Whole genome map of ORFV-1V. (**B**) Whole genome map of ORFV-2W.

**Figure 2 vetsci-11-00617-f002:**
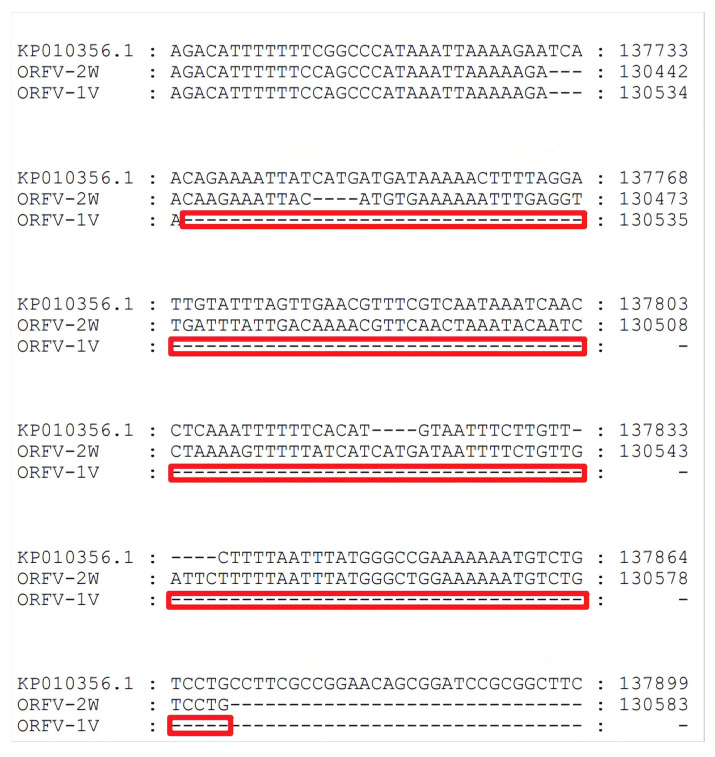
Whole genome nucleotide sequence alignment analysis. The deleted region of ORFV-1V is represented by the red box.

**Figure 3 vetsci-11-00617-f003:**
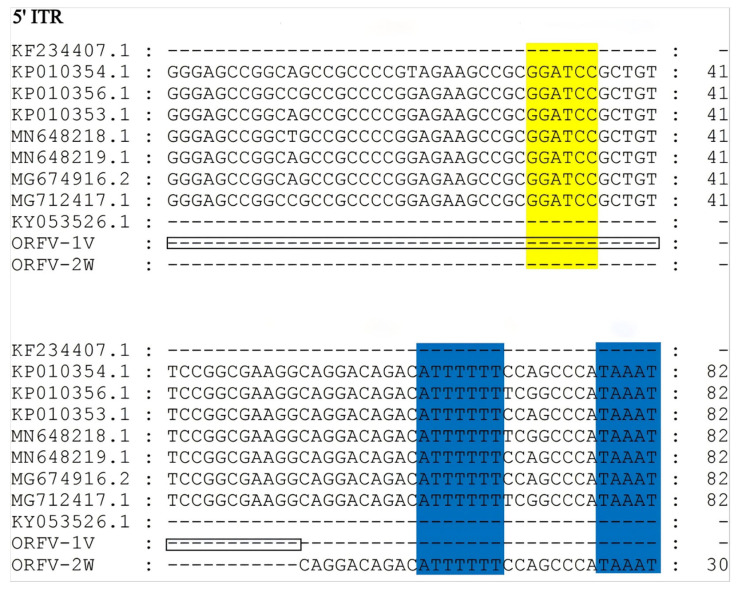
Alignment of ITR sequences from the ORFV-1V and ORFV-2W genomes with those from other isolates. The left terminal 5’-ITR sequences of eleven ORFV isolates were compared using MEGA version 11. The BamHI terminal region (GGATCC) and the telomere resolution sequence (ATT TTTT-N(8)-TAAAT) are indicated as yellow and blue boxes, respectively. The deletion region of ORFV-1V is indicated as a black box.

**Figure 4 vetsci-11-00617-f004:**
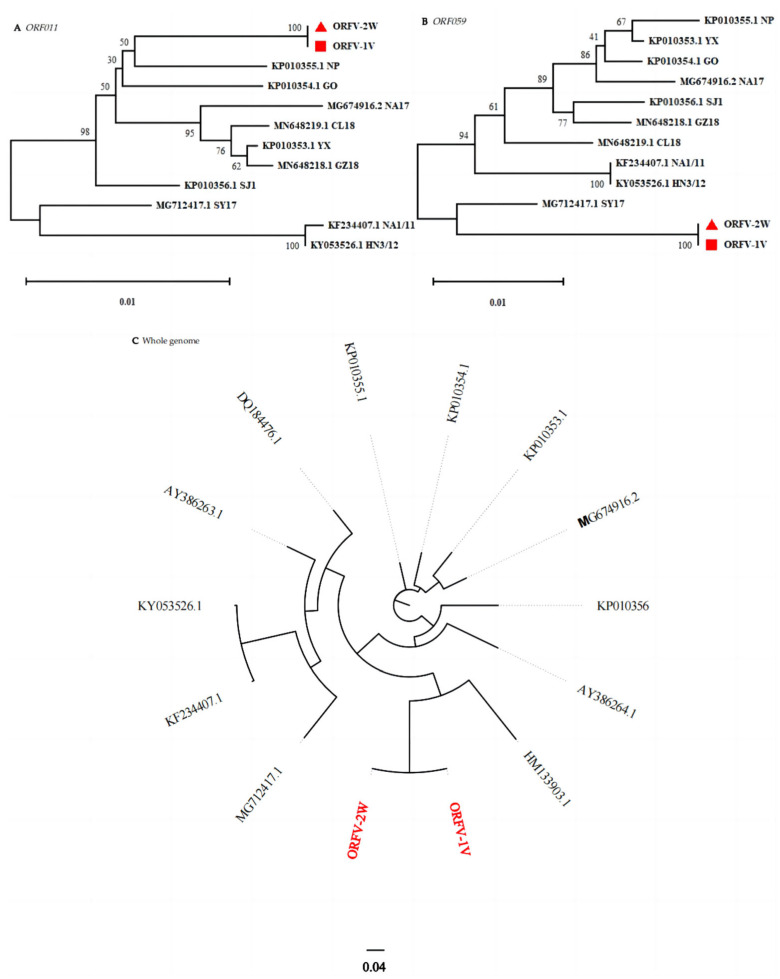
Phylogenetic analysis based on two genes (i.e., ORF011 and ORF059) and whole genome sequence data. Phylogenetic trees were constructed using the neighbor-joining method in MEGA version 11. Numbers above or below branch points indicate the bootstrap support calculated for 1000 replicates. Shown are trees using sequence data for: (**A**) B2L (*ORF011*) and (**B**) F1L (*ORF059*). Here, ORFV-1V and ORFV-2W are represented by red triangles and red squares, respectively. Also shown is a tree using (**C**) whole genome sequence data. ORFV-1V and ORFV-2W are shown in red font.

**Figure 5 vetsci-11-00617-f005:**
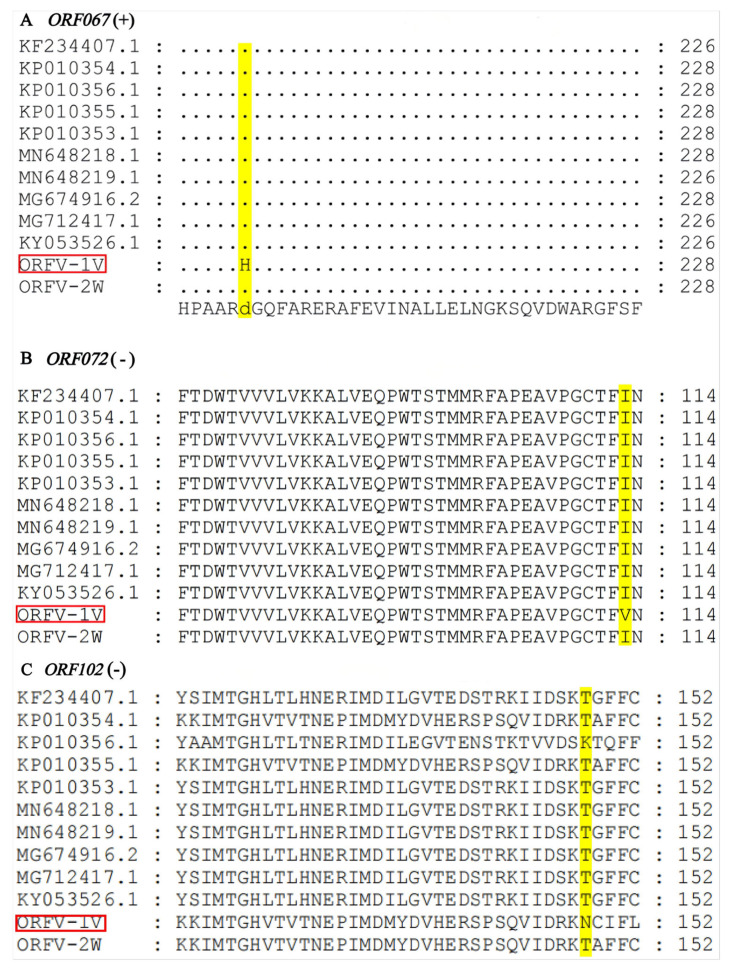
Alignment of amino acid sequences of *ORF067*, *ORF072*, and *ORF102*. The sequences of ORFV-1V and other isolates were aligned using MEGA version 11. (**A**) *ORF067*. (**B**) *ORF072*. (**C**) *ORF102*. ORFV-1V is highlighted with a red box for emphasis. Amino acid mutation sites are represented by yellow boxes.

**Figure 6 vetsci-11-00617-f006:**
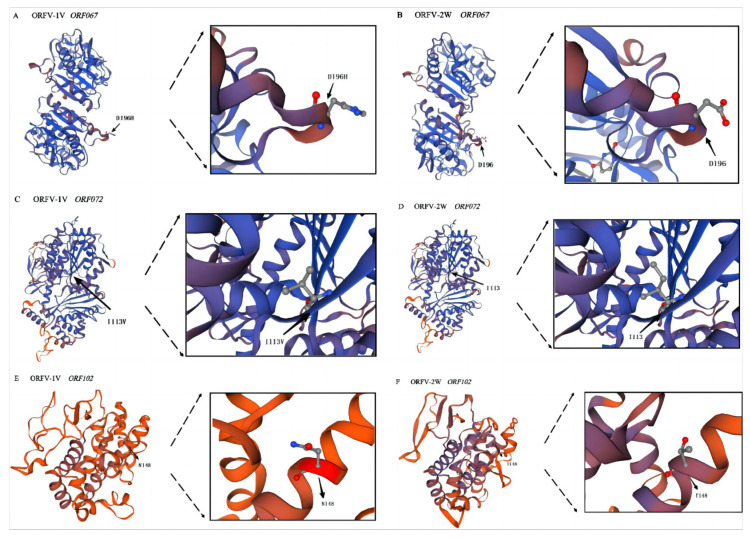
Prediction of the tertiary structures of proteins encoded by *ORF067*, *ORF072*, and *ORF102* of the ORFV-1V and ORFV-2W genomes. Tertiary protein structures were predicted using SWISS-MODEL. The template coverage of each of the above proteins is >50%, and all GMQE > 0.2, which indicates that the prediction results are reliable. (**A**) ORFV-1V *ORF067*. (**B**) ORFV-2W *ORF067*. (**C**) ORFV-1V *ORF072*. (**D**) ORFV-2W *ORF072*. (**E**) ORFV-1V *ORF102*. (**F**) ORFV-2W *ORF102*. Amino acid sites are indicated by black arrows. D196H indicates the mutation of amino acid 196 from aspartic acid (Asp) to histidine (His). I113V shows the mutation of isoleucine (Ile) to valine (Val) at position 113. N148 shows the mutation of amino acid 148 to asparagine (Asn).

**Table 1 vetsci-11-00617-t001:** Summary of complete genomic sequence data of 14 ORFV strains.

Strain Designation	Country	Year	Genome Size (bp)	Host Species	GenBank Accession No.	References
NA1/11	China	2013	137,080	sheep	KF234407.1	[13]
G0	China	2014	139,886	goat	KP010354.1	[14]
SJ1	China	2014	139,112	goat	KP010356.1	[14]
NP	China	2014	132,111	goat	KP010355.1	[14]
YX	China	2014	138,231	goat	KP010353.1	[14]
GZ18	China	2019	138,446	goat	MN648218.1	[15]
CL18	China	2019	138,495	sheep	MN648219.1	[15]
NA17	China	2017	139,287	goat	MG674916.2	[16]
SY17	China	2017	140,413	sheep	MG712417.1	[16]
HN3/12	China	2016	136,643	sheep	KY053526.1	[17]
OV-SA00	USA	2004	139,962	goat	AY386264.1	[18]
D1701	USA	2011	134,038	sheep	HM133903.1	[19]
OV-IA82	USA	2004	137,241	sheep	AY386263.1	[18]
NZ2	New Zealand	2006	137,820	sheep	DQ184476.1	[20]

## Data Availability

The authors confirm that the data supporting the findings of this study are available within the article.

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
