# Peer review of "Genomic Differences and Mutations in Epidemic Orf Virus and Vaccine Strains: Implications for Improving Orf Virus Vaccines"

_vetsci, 2024, doi:10.3390/vetsci11120617_

Round 1
Reviewer 1 Report
Comments and Suggestions for Authors
Dengshuai Zhao et al. provide a manuscript of “Genomic Differences and Mutations in Orf
Virus Epidemic and Vaccine Strain: Implications for Improving Orf Virus Vaccines”. The article conduct a comparative genomic analysis between an epidemic strain (ORFV-2W) and a vaccine strain (ORFV-1V) of Orf Virus (ORFV). Through multiple sequence alignments, phylogenetic tree construction, and single nucleotide polymorphism (SNP) analysis, subtle differences between the two strains were revealed. The study identified missense and frameshift mutations in the ORF067, ORF072, and ORF102 genes of ORFV-1V, as well as deletions in non-coding regions. These discoveries provide new insights into the attenuation mechanism of the vaccine strain.Based on the ORF067, ORF072 and ORF102 genes, SWISS-MODEL were Used to predict the tertiary structures of key proteins in ORFV-1V, revealing the impact of mutations on protein conformations. Although key gene mutations were identified, functional validation of these mutations was not conducted in this study.
Figure 1 as the results should not in the material and method part, in my opinion, the figure 1 do not need in this paper.
Line 91 repeat word “long” ,should be deleted.
Comments on the Quality of English Language
The English is good enough for publish.
Author Response
Responses to reviewers' comments on manuscript No. vetsci-3276087
Dear Reviewers and Editors:
We appreciate all of your comments and suggestions which have been instrumental in enhancing the quality of our manuscript, we are confident that we have addressed each reviewer’s comment as detailed below. The original reviewers’ comments are presented in black and the authors’ responses are highlighted in blue.
Reviewer 1
Dengshuai Zhao et al. provide a manuscript of “Genomic Differences and Mutations in Orf Virus Epidemic and Vaccine Strain: Implications for Improving Orf Virus Vaccines”. The article conduct a comparative genomic analysis between an epidemic strain (ORFV-2W) and a vaccine strain (ORFV-1V) of Orf Virus (ORFV). Through multiple sequence alignments, phylogenetic tree construction, and single nucleotide polymorphism (SNP) analysis, subtle differences between the two strains were revealed. The study identified missense and frameshift mutations in the ORF067, ORF072, and ORF102 genes of ORFV-1V, as well as deletions in non-coding regions. These discoveries provide new insights into the attenuation mechanism of the vaccine strain. Based on the ORF067, ORF072 and ORF102 genes, SWISS-MODEL were Used to predict the tertiary structures of key proteins in ORFV-1V, revealing the impact of mutations on protein conformations. Although key gene mutations were identified, functional validation of these mutations was not conducted in this study.
Answer: Thank you for your suggestions. This manuscript primarily focuses on bioinformatics analysis to discern sequence differences between vaccine strain and epidemic strain. In the subsequent publication, we will validate these differential sites through systematic biological experiments, serving as a complement to this study.
Figure 1 as the results should not in the material and method part, in my opinion, the figure 1 do not need in this paper.
Answer: We agree with your opinion. We have removed Figure 1 in the Materials and Methods section. (Lines 85-87)
Line 91 repeat word “long”, should be deleted.
Answer: Thank you for your comments. We have removed the repeated word in the revised manuscript. (Line 93)
Reviewer 2 Report
Comments and Suggestions for Authors
The Editor Veterinary Sciences
Thank you for the opportunity to review the manuscript: “Genomic Differences and Mutations in Orf Virus Epidemic and 2 Vaccine Strain: Implications for Improving Orf Virus Vaccines”. The paper has been carefully reviewed but significant concerns arose:
Line 39: Contradictory information: Or is a common zoonotic disease, or is rarely seen in humans.
This research deals with the molecular comparison between strains (pathogenic and vaccine) of the ORF virus, inferring, based on genetic/molecular characteristics, its protective capacity against a field challenge. Considering that this work used Chinese viral strains (vaccine and pathogenic sample) , and the ORF virus circulates worldwide, this work presents local results, as there are ORF variants distributed globally, possibly with distinct genetic/molecular characteristics.
The work has merits. However, literature describes that this relationship between genetic/molecular characteristics is not 100% equivalent when evaluated in vivo. Therefore, cell culture assays, or even experimental infections, could provide more support to the final results obtained.
The work has great importance, presenting important scientific merits, with regional information, but results are overestimated. The discussion is very long and difficult to understand, without objectivity. It needs to be adjusted. It is recognized that in vitro studies may not correspond with studies performed in vivo.The work should be reduced and rewritten objectively.
Author Response
Responses to reviewers' comments on manuscript No. vetsci-3276087
Dear Reviewers and Editors:
We appreciate all of your comments and suggestions which have been instrumental in enhancing the quality of our manuscript, we are confident that we have addressed each reviewer’s comment as detailed below. The original reviewers’ comments are presented in black and the authors’ responses are highlighted in blue.
Reviewer 2
The Editor Veterinary Sciences
Thank you for the opportunity to review the manuscript: “Genomic Differences and Mutations in Orf Virus Epidemic and 2 Vaccine Strain: Implications for Improving Orf Virus Vaccines”. The paper has been carefully reviewed but significant concerns arose:
Line 39: Contradictory information: Or is a common zoonotic disease, or is rarely seen in humans.
Answer: Thank you for your comments. To clarify, we have revised the text to accurately convey. (Line 39)
This research deals with the molecular comparison between strains (pathogenic and vaccine) of the ORF virus, inferring, based on genetic/molecular characteristics, its protective capacity against a field challenge. Considering that this work used Chinese viral strains (vaccine and pathogenic sample), and the ORF virus circulates worldwide, this work presents local results, as there are ORF variants distributed globally, possibly with distinct genetic/molecular characteristics.
Answer: Thank you for your comments. We appreciate your concerns regarding the emphasis on local strains. However, we believe that the molecular comparison of these strains is essential for understanding the overall behavior of the ORF virus. Studying these specific strains offers valuable insights into their genetic characteristics, which may influence vaccine efficacy. We also acknowledge the limitations and have provided clarifications in the discussion to highlight their significance in the context of China. (Lines 303-322)
The work has merits. However, literature describes that this relationship between genetic/molecular characteristics is not 100% equivalent when evaluated in vivo. Therefore, cell culture assays, or even experimental infections, could provide more support to the final results obtained.
Answer: Thank you for your suggestions. Definitely the genetic/molecular characteristics is not 100% equivalent when evaluated in vivo, nevertheless, this paper concentrates on bioinformatics to identify sequence variations between vaccine and epidemic strains. Future work will experimentally confirm these variations, including in vivo and in vitro studies,we will present these data and results in next article.
The work has great importance, presenting important scientific merits, with regional information, but results are overestimated. The discussion is very long and difficult to understand, without objectivity. It needs to be adjusted. It is recognized that in vitro studies may not correspond with studies performed in vivo. The work should be reduced and rewritten objectively.
Answer: Thank you very much for your comments. We have revised it to ensure that our conclusions are grounded in the data and presented objectively. We have also streamlined the discussion section to enhance clarity and focus on the key findings.
Reviewer 3 Report
Comments and Suggestions for Authors
General observations:
The submitted manuscript presents an interesting comparative work between an endemic strain of the Orf virus and an attenuated vaccine used to protect animals from the effects of the endemic virus.
The authors sequenced the complete genome of the two viruses, presenting robust evidence of a close phylogenetic relationship between these two viruses compared to other field strains of Orf virus. They performed a thorough SNP analysis, revealing interesting gene mutations. They then focused on these mutations showing differences, especially at the level of the ORF067, ORF072, and ORF102 genes and in the terminal non-coding regions.
The manuscript is well-written and enjoyable to read. The differences between the genomes, which in part affect the encoded proteins' tertiary structure, suggest several potential attenuation mechanisms. These observations are essential, especially in view of the future development of attenuated vaccines produced by inserting targeted mutations into the genomes of virulent viruses. This point is the strength of this work and, unfortunately, its weakness since the authors can only present educated guesses without functional data. Future work with a molecular clone of ORFV-2W and ORFV-1V will permit the authors to reveal these functional details.
Minor points:
Lines 30 and 50: Pathogenicity and virulence appear pleonastic; please select one.
Line 49: viral replication and reproduction; I would delete reproduction and keep only replication.
Author Response
Responses to reviewers' comments on manuscript No. vetsci-3276087
Dear Reviewers and Editors:
We appreciate all of your comments and suggestions which have been instrumental in enhancing the quality of our manuscript, we are confident that we have addressed each reviewer’s comment as detailed below. The original reviewers’ comments are presented in black and the authors’ responses are highlighted in blue.
Reviewer 3
General observations:
The submitted manuscript presents an interesting comparative work between an endemic strain of the Orf virus and an attenuated vaccine used to protect animals from the effects of the endemic virus.
The authors sequenced the complete genome of the two viruses, presenting robust evidence of a close phylogenetic relationship between these two viruses compared to other field strains of Orf virus. They performed a thorough SNP analysis, revealing interesting gene mutations. They then focused on these mutations showing differences, especially at the level of the ORF067, ORF072, and ORF102 genes and in the terminal non-coding regions.
Answer: Thank you very much for your positive comments. We appreciate your recognition of the comparative work we have conducted between the endemic strain of the ORF virus and the attenuated vaccine.
The manuscript is well-written and enjoyable to read. The differences between the genomes, which in part affect the encoded proteins' tertiary structure, suggest several potential attenuation mechanisms. These observations are essential, especially in view of the future development of attenuated vaccines produced by inserting targeted mutations into the genomes of virulent viruses. This point is the strength of this work and, unfortunately, its weakness since the authors can only present educated guesses without functional data. Future work with a molecular clone of ORFV-2W and ORFV-1V will permit the authors to reveal these functional details.
Answer: Thank you very much for your positive comments. We agree that future research using molecular clones of ORFV-2W and ORFV-1V will be essential for validating these hypotheses and uncovering their functional details. We will conduct further investigations in our future work.
Minor points:
Lines 30 and 50: Pathogenicity and virulence appear pleonastic; please select one.
Answer: Thank you for your comments. We have revised the manuscript to use only one of these terms for clarity. (Lines 30 and 51)
Line 49: viral replication and reproduction; I would delete reproduction and keep only replication.
Answer: Thank you for your comments. We have deleted the reproduction word in the revised manuscript. (Line 50)